# Methylene Blue Inhibits Cromakalim-Activated K^+^ Currents in Follicle-Enclosed Oocytes

**DOI:** 10.3390/membranes13020121

**Published:** 2023-01-18

**Authors:** Dmytro Isaev, Keun-Hang Susan Yang, Georg Petroianu, Dietrich Ernst Lorke, Murat Oz

**Affiliations:** 1Department of Cellular Membranology, Bogomoletz Institute of Physiology, 01024 Kiev, Ukraine; 2Department of Biological Sciences, Schmid College of Science and Technology, Chapman University, One University Drive, Orange, CA 92866, USA; 3Department of Pharmacology, College of Medicine and Health Sciences, Khalifa University, Abu Dhabi 127788, United Arab Emirates; 4Department of Anatomy and Cellular Biology, College of Medicine and Health Sciences, Khalifa University, Abu Dhabi 127788, United Arab Emirates; 5Department of Pharmacology and Therapeutics, Faculty of Pharmacy, Kuwait University, Safat 13110, Kuwait

**Keywords:** cromakalim, follicular cells, K_ATP_ channels, methylene blue, *Xenopus* oocyte

## Abstract

The effects of methylene blue (MB) on cromakalim-induced K^+^ currents were investigated in follicle-enclosed *Xenopus* oocytes. In concentrations ranging from 3–300 μM, MB inhibited K^+^ currents (IC_50_: 22.4 μM) activated by cromakalim, which activates K_ATP_ channels. MB inhibited cromakalim-activated K^+^ currents in a noncompetitive and voltage-independent manner. The respective EC_50_ and slope values for cromakalim-activation of K^+^ currents were 194 ± 21 µM and 0.91 for controls, and 206 ± 24 µM and 0.87 in the presence of 30 μM MB. The inhibition of cromakalim-induced K^+^ currents by MB was not altered by pretreatment with the Ca^2+^ chelator BAPTA, which suggests that MB does not influence Ca^2+^-activated second messenger pathways. K^+^ currents mediated through a C-terminally deleted form of Kir6.2 (KirΔC26), which does not contain the sulfonylurea receptor, were still inhibited by MB, indicating direct interaction of MB with the channel-forming Kir6.2 subunit. The binding characteristics of the K_ATP_ ligand [^3^H]glibenclamide are not altered by MB in a concentration range between 1 μM-1 mM, as suggested by radioligand binding assay. The presence of a membrane permeable cGMP analogue (8-Br-cGMP, 100 µM) and a guanylate cyclase activator (BAY 58-2667, 3 µM) did not affect the inhibitory effects of MB, suggesting that MB does not inhibit cromakalim-activated K^+^ currents through guanylate cyclase. Collectively, these results suggest that MB directly inhibits cromakalim-activated K^+^ currents in follicular cells of *Xenopus* oocytes.

## 1. Introduction

Methylene Blue (MB), a cationic dye belonging to the phenothiazine-class, is utilized in several clinical conditions, e.g., treatment of methemoglobinemia and malaria, and has been suggested to have beneficial effects in the management of pain, depression, and neurodegenerative diseases [1,2,3]. One of the most established pharmacological actions of MB is its vasoconstrictive effect in various vascular beds. MB induces vasoconstriction in both preclinical [4,5] and clinical studies [6,7]. Thus, vasoconstrictive actions of MB are utilized currently to elevate blood pressure in numerous critical clinical conditions, e.g., cardiovascular and septic shock, and cardioplegia [8]. However, the mechanisms mediating these therapeutic effects remain largely unknown. It is assumed that inhibition of the cGMP pathway is the main mechanism underlying the vasoconstrictive actions of MB. However, several studies suggest that MB has also cGMP-independent direct actions on ion channels [9,10,11], neurotransmitter receptors [12,13], transporters [14], and enzymes [15,16]. 

K_ATP_ (ATP-sensitive potassium) channels are involved in the regulation of vasoconstriction, heart rhythm and insulin secretion. Low intracellular ATP levels or K_ATP_ channel openers, e.g., cromakalim, pinacidil, diazoxide [17,18], activate K_ATP_ channels; while glibenclamide and other antidiabetic sulfonylureas suppress their activity [19,20,21]. A broad range of diseases, e.g., diabetes, cardiac arrhythmias, hypertension, is related to their dysfunction [22,23,24]. K_ATP_ channels play an important role in controlling vasoconstriction [22,23,25]; it is therefore likely that their modulation may contribute to some of the previously described vasoconstrictive actions of MB.

*Xenopus laevis* oocytes are surrounded by follicular cells expressing endogenous K_ATP_ channels and are connected to oocytes through gap-junctions [26,27,28]. These channels can be utilized to characterize the influence of pharmacological compounds upon K_ATP_ channels, even though they may not exhibit exactly the same pharmacological characteristics as those of pancreatic β-cells or of vascular beds. In this manuscript, using two-electrode voltage clamp techniques on follicle-enclosed oocytes, we describe how MB modifies the function of K_ATP_ channels.

## 2. Materials and Methods

Mature female *Xenopus laevis* frogs (Xenopus I, Ann Arbor, MI, USA) were kept in dechlorinated tap water at 19–21 °C and fed with beef liver at least twice a week. Clusters of oocytes were removed surgically under tricaine (Sigma, St. Louis, MO, USA) anesthesia (0.15%). Individual oocytes were manually dissected in a solution containing (in mM): NaCl, 88; KCl, 1; NaHCO_3_, 2.4; MgSO_4_, 0.8; HEPES, 10 (pH 7.5) and stored 2–7 days in modified Barth’s solution containing (in mM): NaCl, 88; KCl, 1; NaHCO_3_, 2.4; Ca(NO_3_)_2_, 0.3; CaCl_2_, 0.9; MgSO_4_, 0.8; HEPES, 10 (pH 7.5), supplemented with sodium pyruvate 2 mM, penicillin 10,000 IU/L, streptomycin 10 mg/L, and gentamicin 50 mg/L [29,30]. Oocytes were placed in a 0.2 mL recording chamber and superfused at a constant rate of 5–7 mL/min. The bathing solution consisted of (in mM): NaCl, 95; KCl, 2; CaCl_2_, 2; and HEPES 5 (pH 7.5). The cells were impaled at the animal pole with two glass microelectrodes filled with 3 M KCl (1–3 MΩ). The oocytes were routinely voltage clamped at a holding potential of −20 mV using a GeneClamp-500 amplifier (Axon Instruments Inc., Burligame, CA, USA). Current responses were digitized by an A/D converter and analyzed using pClamp 6 (Axon Instruments Inc., Burligame, CA, USA) run on an IBM/PC or directly recorded on a Gould 2400 rectilinear pen recorder (Gould Inc., Cleveland, OH, USA). Current-voltage characteristics were studied using 1 s voltage steps (–120 to 20 mV). All chemicals and drugs including glibenclamide, MB HCl, BAPTA-AM, and 8-Br-cAMP were from Sigma (St. Louis, MO, USA). BAY 58-2667 (dissolved in DMSO) was obtained from Tocris-Bio-Techne (Minneapolis, MN, USA). Drugs were applied externally by addition to the superfusate. Procedures for the injections of BAPTA (50–70 nL, 100 mM) were described earlier in detail [31].

For inside-out patch experiments, oocytes were chemically defolliculated by collagenase 1A treatment (Sigma, St. Louis, MO, USA; 2 mg/mL, 2 h) and injected with 2 ng cRNA encoding Kir6.2ΔC26 mutant. After 2 days of incubation in modified Barth’s solution, oocytes were placed in hypertonic solution containing 200 mM K^+^-aspartate at pH 7.0, and the vitelline layer was removed with sharpened watchmaker’s forceps. The pipette (external) solution contained (in mM) 140 KCl, 1.2 MgCl_2_, 2.6 CaCl_2_, and 10 HEPES (pH 7.4). Intracellular (bath) solution contained (in mM) 107 KCl, 10 EGTA, 2 MgCl_2_, 1 CaCl_2_, and 10 HEPES (pH 7.2 with KOH; final K^+^ ≅140). Patch pipettes had a resistance of 250–400 kΩ when filled with the pipette solution. Currents were recorded at 20–22 °C from giant inside-out patches using Axopatch 200B amplifier (Axon Instruments Inc., Burligame, CA, USA) at a holding potential of 0 mV, sampled at a rate of 2 kHz and filtered at 1 kHz. Currents were evoked by 2 s voltage ramps from −100 mV to +100 mV at a pulse frequency of 0.5 Hz. In each inside-out patch, the efficacy of 1 mM K_2_ATP to block the K_ATP_ current was tested before applying MB. Leak currents recorded at 10 mM K_2_ATP at the end of the experiments were subtracted from recordings. The recording chamber had a volume of 250 μL and the perfusion rate was 2 mL/min. Due to light sensitivity of MB, experiments were conducted in the dark and MB containers and perfusion lines kept in the dark by aluminum foils.

Statistical significance at the level of 0.05 was analyzed using the Student’s *t*-test, paired *t*-test or ANOVA. Concentration-response curves were obtained by fitting the data to the logistic equation,
y = E_max_/(1 + [x/EC_50_]^n^) 
where x and y are concentration and response, respectively, E_max_ is the maximal response, EC_50_ is the half-maximal concentration, and n is the slope factor. For data analysis, calculations, and fits of the data, the computer software Origin 8.5 (Microcal Software-OriginLab Corp., Northampton, MA, USA) was used. 

For radioligand binding experiments, follicle-enclosed oocytes were suspended in 300 mL of buffer containing 50 mM HEPES, 300 mM sucrose and 1 mM EDTA at 4 °C on ice as described earlier [32,33]. Oocytes were homogenized using a motorized Teflon homogenizer (six strokes, 15 s each at high speed). This was followed by sequential centrifugations at 1000× *g* (10 min) and 10,000× *g* (20 min); each time the pellet was discarded and the supernatant was used for the subsequent step. The final centrifugation was at 60,000× *g* for 25 min. The microsomal pellet, which contains the membranes of follicular cells [34], was resuspended in 50 mM HEPES buffer and used for the binding studies. 

The radioligand binding experiments were carried out at room temperature (20–22 °C) for 1 h. Oocyte membranes were incubated in 1 mL of 50 mM HEPES, pH 7.5, at a protein concentration of 200–500 μg/mL. [^3^H]glibenclamide was dissolved in ethanol/dimethyl sulphoxide (1:1). For each experiment, freshly made glibenclamide solution was used. For the analysis, calculations, nonlinear curve fitting and regression fits of the radioligand binding data, the computer software Origin 8.5 (Microcal Software-OriginLab Corp., Northampton, MA, USA) was used. 

## 3. Results

A bath application of 100 μM cromakalim for 5 min changed the mean value for the resting membrane potential from –38 ± 4 mV (mean ± SE, *n* = 12) to –85 ± 5 mV (*n* = 12), which is close to the reversal potential for K^+^ in oocytes [35]. In line with earlier investigations [26,27,36], bath application of cromakalim activated a slowly developing outward current (Figure 1A) and maximal amplitudes of these currents did not change significantly during consecutive administrations of cromakalim every 10 min for up to 80 min. Glibenclamide (1 μM), a specific blocker of K_ATP_ channels, reversibly inhibited cromakalim-activated current (51 ± 4% inhibition; data not shown, *n* = 7), and the directions of these currents reversed by increasing the external K^+^ concentration to 100 mM (data not shown, *n* = 4).

The administration of MB (30 μM) for 20 min caused a significant inhibition of the slow-outward current induced by 100 μM cromakalim with incomplete recovery during the washout period (Figure 1B). Increasing the MB administration time to 30 min (*n* = 3) did not cause further inhibition, suggesting that the effect of MB reached a steady state within 10 to 20 min. Figure 1C shows concentration-dependent inhibition of cromakalim-induced outward current by MB. The minimum concentration of MB causing a significant inhibition of outward current was 3 μM (11 ± 4% inhibition; *n* = 4–5; *p <* 0.05, paired *t*-test). The IC_50_ (a fifty percent of maximal MB inhibition) and slope value (*n*) were 22.4 μM and 1.1, respectively, and the maximum inhibition was reached at the concentrations of 300 μM and above (84 ± 5% inhibition, *n* = 5–7). 

The current-voltage (I-V) relationship of cromakalim-induced currents in the absence and presence of MB (30 µM) is presented in Figure 2A. The effect of MB on the cromakalim-induced net outward current (cromakalim-activated current minus resting current at given voltage) did not show voltage dependence i.e., the extent of MB inhibition was not changed significantly in the voltage range studied (Figure 2B). In addition, in the absence and presence of 30 μM MB, the reversal potentials of the cromakalim-activated currents were –89 ± 4 mV and –92 ± 3 mV, respectively (*p* > 0.05, paired *t*-test; *n* = 5), indicating MB did not alter the ionic selectivity of the K_ATP_ channels.

Cell bodies of oocytes are coupled to follicular cells through gap junctions (for reviews, [28,35]), therefore, MB may affect gap junctions and alter the resistance of the ionic pathways. For this reason, the resistances in follicle-enclosed (to investigate the involvement of R_o_) and enzymatically defolliculated oocytes (to investigate the involvements of R_j_ and R_f_) in the presence and absence of MB were measured (without involvement of cromakalim-induced conductances in follicular cells). There was no significant change in resistances measured from defolliculated or follicle-enclosed oocytes, in the presence and absence of MB (*n* = 14–16; Student’s *t*-test, *p* > 0.05, Figure 2C).

MB has been demonstrated to cause changes in intracellular Ca^2+^ homeostasis [37,38,39] and modulate the functions of Ca^2+^-activated K^+^ channels [11,40,41,42]. Thus, the effects of MB on Ca^2+^-dependent second messenger systems or Ca^2+^-activated K^+^ and/or Cl^−^ channels may interfere with the effect of MB on cromakalim-induced currents. In order to investigate the involvement of intracellular Ca^2+^ in the MB effect, follicle-enclosed oocytes were incubated in BAPTA-AM (5 mM) overnight (12-h) and injected with BAPTA (50 nL, 100 mM) 15 min before the recordings to ensure the chelation of intracellular Ca^2+^ in both follicular cells and oocytes. Percent inhibitions of cromakalim-induced currents by MB were not significantly different between BAPTA-treated oocytes and controls injected with 50 nL distilled-water (Figure 2D, *p* > 0.05, Student’s *t*-test; *n* = 5). In addition, current-voltage relationships in the absence and presence of MB indicated no significant alterations in reversal potentials in BAPTA-treated oocytes (−89 ± 3 versus −91 ± 4; *p* > 0.05, Student’s *t*-test; *n* = 5). 

The cromakalim binding site(s) may be involved in MB inhibition of K_ATP_ channels. For this reason, we examined concentration-response curves of cromakalim activation in the absence and presence of MB (30 μM). MB did not cause significant changes in the EC_50_ values but inhibited the maximal cromakalim-activated currents (54 ± 5 % of controls; *n* = 5–6). Respective EC_50_ and slope values in the absence and presence of MB (30 μM) were 194 ± 21 µM and 0.91 vs. 206 ± 24 µM and 0.87, suggesting that MB inhibition of cromakalim-induced K^+^ currents occurs in a noncompetitive manner.

The K_ATP_ channel is comprised of four Kir6.2 subunits and each subunit is associated with a larger regulatory sulfonylurea receptor (SUR) subunit (for a review, [43]). Therefore, we investigated the effect of MB on the specific binding of [^3^H]glibenclamide, a sulfonylurea class drug, in the microsomal fraction of *Xenopus* oocytes. Figure 3B shows equilibrium curves for the binding of [^3^H]glibenclamide, in the absence and presence of MB (30 µM). Maximum bindings (B_max_) of [^3^H]glibenclamide for controls and MB-treated membranes were 5.78 ± 0.39 and 5.54 ± 0.41 pmol/mg, respectively (*p* > 0.05, Student’s *t*-test; *n* = 5). The affinities (K_d_) of [^3^H]glibenclamide for controls and MB-treated membranes were 1.21 ± 0.14 and 1.32 ± 0.22 nM, respectively. Similarly, the specific binding of [^3^H]glibenclamide was not altered by the incubation of microsomal membranes with increasing concentrations (1 μM to 1 mM) of MB (Figure 3C).

The C-terminally truncated form of Kir6.2 (KirΔC26) lacks the last 26 amino acids but forms functional ATP-sensitive channels in the absence of the SUR subunit [44]. This mutant channel was employed to investigate if MB can act on the channel-forming Kir6.2 subunit in the absence of SUR. The application of MB (30 μM) for 40 s caused a significant inhibition of currents mediated through KirΔC26 subunits in a voltage-independent manner (Figure 4A). The reversal potential was not altered in the presence of MB; reversal potentials were 2 ± 3 mV and 0 ± 2 mV in the presence and absence of MB, respectively. MB caused a 48 ± 5% (*n* = 4) inhibition of controls (at −100 mV), and recovery was incomplete within the time course of the experiments (Figure 4B).

MB is a known inhibitor of soluble guanylate cyclase [45,46], and K_ATP_ channels have been shown to be modulated by cGMP [47,48]. We have tested the involvement of guanylate cyclase activity by investigating the effects of BAY 58-2667 (3 µM), a potent activator of soluble guanylate cyclase [49], and 8-Br-cGMP (100 µM), a membrane-permeable cGMP analogue, on the MB inhibition of cromakalim-activated K^+^ currents in follicle-enclosed oocytes. Application of BAY 58-2667 alone or 8-Br-cGMP alone for 20 min. caused 14 ± 4% (*p* < 0.05, *n* = 5, Student’s *t*-test) and 16 ± 4% (*p* < 0.05, *n* = 6, Student’s *t*-test) potentiation of cromakalim-activated K^+^ currents, respectively. However, after 20 min preincubation with BAY 58-2667 or 8-Br-cAMP, the extent of MB inhibition was not significantly different from MB alone (*p* > 0.05, *n* = 5–7, Student’s *t*-test; Figure 4C). 

## 4. Discussion

Our results indicate that cromakalim-activated K^+^ currents in follicle-enclosed oocytes were inhibited by MB in a non-competitive (with respect to cromakalim and glibenclamide binding sites) and cGMP-independent manner. Follicular cells are electrically coupled to oocytes through gap junctions (for reviews, [28,35]), and, thus, an effect of MB on gap junctions would alter membrane resistance (through oocyte, gap junctions, and follicular cells) and interfere with MB inhibition of K^+^ currents. MB, however, did not cause a significant alteration in the cell input resistances in either follicle-enclosed or defolliculated oocytes, suggesting that, when cromakalim-activated channels are closed, other ionic conductances are not significantly affected by MB in either defolliculated or follicle-enclosed oocytes.

Changes in the intracellular Ca^2+^ levels would alter the activities of Ca^2+^-activated Cl^−^ and K^+^ channels and, thus, could interfere with the effect of MB on cromakalim-activated K^+^ currents. MB inhibited the cromakalim-activated K^+^ currents in BAPTA-treated oocytes to the same extent as in untreated oocytes. Similarly, the reversal potential of the cromakalim-induced current was not changed significantly, suggesting that Ca^2+^-activated Cl^−^ conductances are not significantly involved in the effect of MB on K^+^ currents. In addition, since the cromakalim-activated currents were recorded near the reversal potential (−20 mV) for Ca^2+^-activated Cl^−^ channels in oocytes [50], Cl^−^ currents are not likely to interfere with the effect of MB on K_ATP_ channels. In line with earlier investigations [26], the current-voltage relationship for cromakalim-activated currents was linear within the voltage range studied (−120 to 20 mV); neither the linear characteristics nor the reversal potential for cromakalim-activated K^+^ currents was altered by MB (Figure 2A) and MB inhibition of cromakalim-activated K^+^ currents was voltage-independent (Figure 2B).

Analysis of cromakalim concentration-response curves in the presence and absence of MB indicated that MB inhibits K^+^-currents in a non-competitive manner, suggesting that MB does not act by inhibiting the cromakalim binding site on the channel. Similarly, radioligand binding studies with [^3^H]glibenclamide, a sulfonylurea-class antidiabetic drug, showed that MB does not significantly affect the glibenclamide binding site in follicle-enclosed oocyte membranes either. The pore of the K_ATP_ channel is formed from four Kir6.2 subunits, each of which is associated with a larger regulatory sulfonylurea receptor (SUR) subunit, which is the primary target for K_ATP_ blockers [43]. Importantly, MB also inhibited ion currents in oocytes expressing the C-terminally deleted mutant of Kir6.2 (KirΔC26), which forms functional channels but does not contain a sulfonylurea binding site (SUR) [44]. Thus, the results of electrophysiological studies on KirΔC26 mutant channel and radioligand binding studies suggest that MB does not suppress the function of K_ATP_ channels by inhibiting the known sulfonylurea binding site, but rather by acting at a different site on the K_ATP_ channel. Collectively, these results indicate that neither cromakalim nor sulfonylurea binding sites are involved in the inhibition of K_ATP_ channels by MB.

Similar to our findings, MB has been suggested to have direct inhibitory effects on the functions of K^+^ [11,40,41,42] and Na^+^ [9] channels and α_7_ nicotinic receptors [13] with IC_50_ values ranging from 10 to 100 μM (for a review, [3]). MB is a phenothiazine-based molecule, and structurally similar compounds, such as promethazine [51], chlorpromazine [27,52] mefloquine, quinine, quinidine, and quinacrine [53,54], have been shown to directly inhibit K_ATP_ channels. Collectively, these earlier results support our findings suggesting that MB directly inhibits the function of K_ATP_ channels.

Based on our findings, we speculate that some of the cardiovascular effects of MB are mediated by inhibition of K_ATP_ channels. In previous investigations, MB, in concentration ranges used in the present study, was reported to induce depolarizations in various cell types in a cGMP-independent manner [40,55,56,57]. Depolarization by suppression of K_ATP_ channels has been demonstrated to cause activation of voltage-gated Ca^2+^ channels and to increase intracellular Ca^2+^ levels, and a subsequent vasoconstriction in vascular beds (for reviews, [17,22]). Therefore, the suppression of K_ATP_ channels may cause depolarizations of the cell, activate voltage-gated Ca^2+^ channels, and subsequently contribute to MB-induced vasoconstriction. In conclusion, our results suggest that inhibition of K_ATP_ channels by MB may be one of the mechanisms contributing to the vasoconstrictive effects of this compound. However, the elucidation of the mechanisms underlying MB’s vasoconstrictive effects will require further studies analyzing the roles of K_ATP_ channels in vasoconstriction of a specific vascular bed.

## Figures and Tables

**Figure 1 membranes-13-00121-f001:**
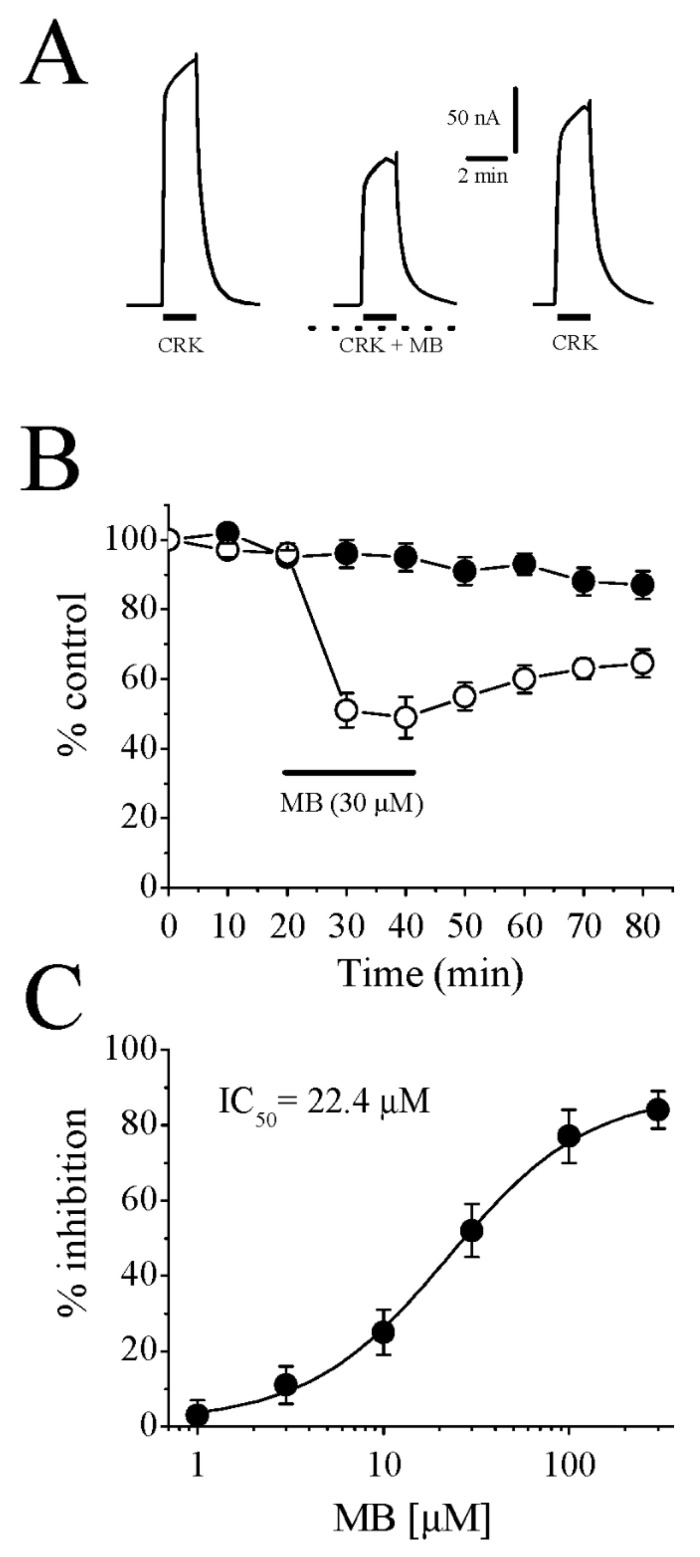
Cromakalim-induced outward currents are inhibited by MB in follicular cells of *Xenopus* oocytes. (**A**) Currents activated by 100 μM cromakalim (*left*), during co-administration of cromakalim and 30 μM MB following 20 min MB preincubation (*middle*), 20 min recovery (*right*). Administration times for cromakalim are shown with solid horizontal bars. Dotted lines represent continuous MB application during recordings. (**B**) Time course of the peak cromakalim-activated currents are shown in the absence (*filled circles*) and the presence of 30 µM MB (*open circles*). Each data point indicates the normalized means and S.E.M. of five to six experiments. The horizontal bar represents the duration of MB administration. (**C**) Concentration-response curve for the inhibitory effect of MB on cromakalim (100 μM)-activated currents. Data points are shown as means ± S.E.M. (*n* = 4–5). The curve is the best fit of the data to the logistic equation presented in the methods section. CKL: cromakalim; MB: methylene blue.

**Figure 2 membranes-13-00121-f002:**
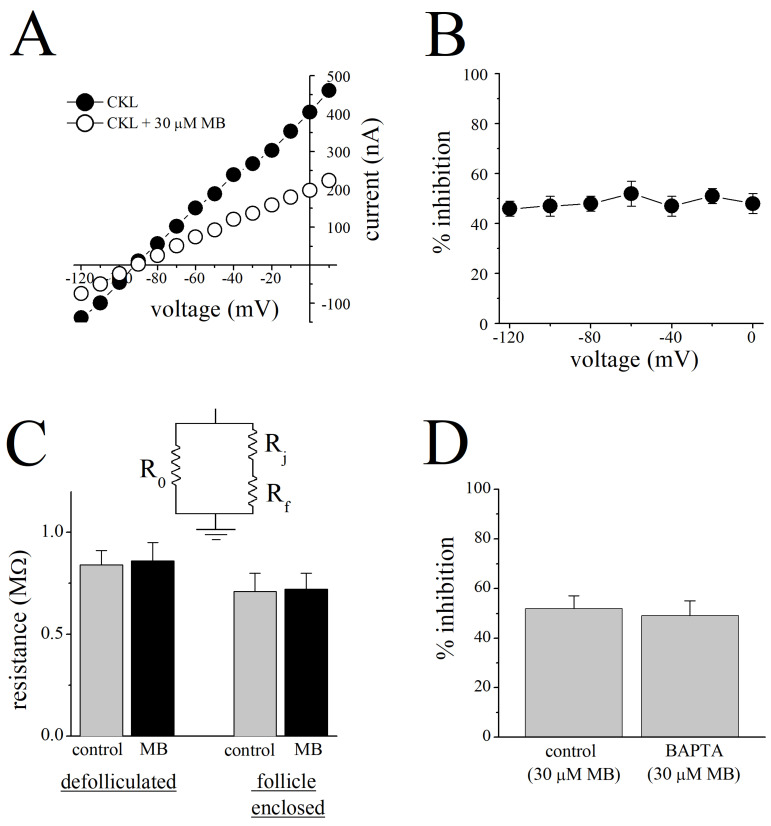
Inhibition of cromakalim-induced K^+^ currents by MB is voltage-independent and not mediated through gap junctions. (**A**) Current-voltage relationship of cromakalim-activated currents recorded during 1 s voltage steps applied before (*filled circles*) and after application of 30 μM MB (*open circles*). (**B**) Different membrane potentials did not alter percentage inhibition of cromakalim-activated K^+^ currents by MB. Differences among the means of current inhibitions by 30 µM MB at different holding potentials were not statistically significant (*p* > 0.05, ANOVA, *n* = 5–7). (**C**) Inset to Figure 2C indicates the equivalent resistive-circuit diagram. The mean values for the sum of resistances through the oocyte gap junction (R_j_), follicular cell membranes (R_f_) in follicle-enclosed oocytes before (*gray bars*) and after 20 min administration of 30 μM MB (*black bars)* are shown on the right side (*n* = 14). Membrane resistance (R_o_) of enzymatically defolliculated oocytes before and after MB treatment are presented on the left (*n* = 16). Values of resistances were calculated from current-voltage curves recorded in the range between –50 mV and +10 mV. (**D**) Effect of MB (30 μM) on cromakalim (100 μM)-induced K^+^ currents in control and in BAPTA-pretreated oocytes. CKL: cromakalim; MB: methylene blue.

**Figure 3 membranes-13-00121-f003:**
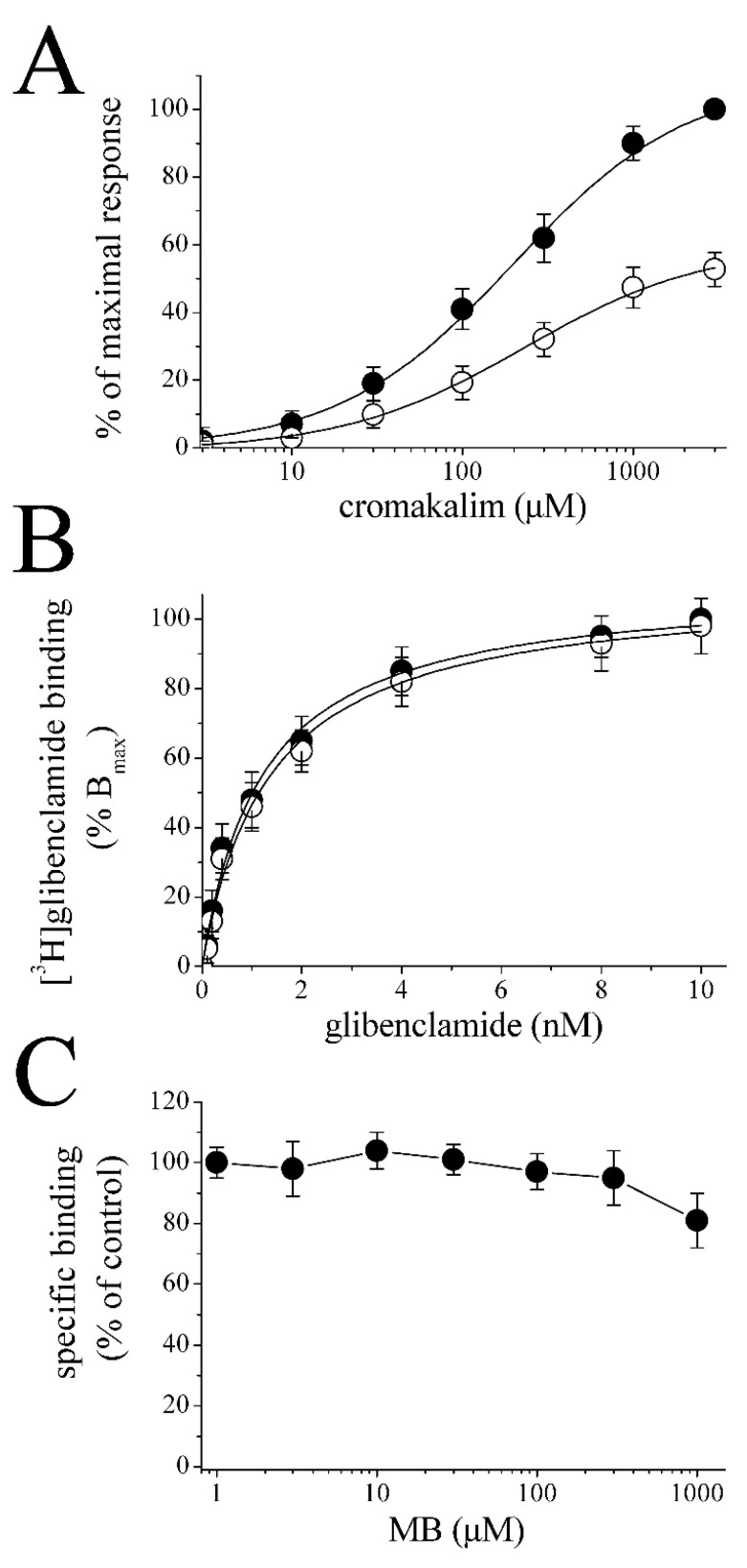
The effects of MB on the cromakalim dose-response curve and the specific binding of [^3^H]glibenclamide. (**A**) Concentration-response curves for cromakalim-activated currents in the absence (*filled circles*) and presence (*open circles*) of MB (30 μM). MB was administered for 20 min, and cromakalim and MB were then co-administered for 2 min. Data points represent the mean ± S.E.M. (*n* = 5–6; error bars not visible are smaller than the size of the symbols). Curves show the best fit of the data to the logistic equation presented in the methods section. The concentration-response curves are normalized to maximal control cromakalim response. (**B**) Specific binding as a function of the concentration of [^3^H]glibenclamide in the absence (*filled circles*) and presence of 30 μM MB (*open circles*). Data represent the means of four experimental measurements. The incubation time was 60 min at 22 °C, pH 7.5. In order to determine nonspecific binding, samples were incubated with 10 nM of unlabeled glibenclamide. (**C**) Increasing concentrations of MB do not alter the specific binding of [^3^H]glibenclamide (1 nM). Data represent the results of 4–5 experiments. Data points show means ± S.E.M. Microsomal membranes were incubated with 1 nM [^3^H]glibenclamide (0.3–0.5 mg/mL for 60 min) with increasing concentrations of MB in the medium. Free and bound [^3^H]glibenclamide were separated by filtration. MB: methylene blue.

**Figure 4 membranes-13-00121-f004:**
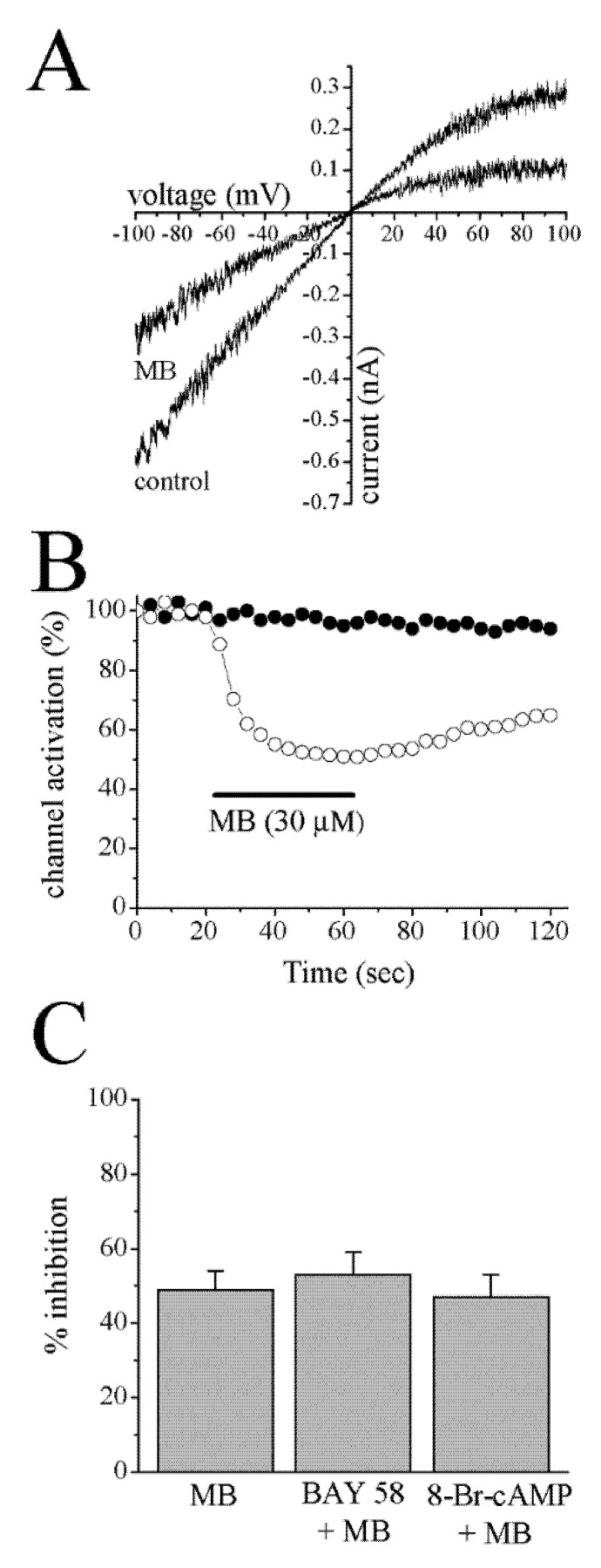
Effects of MB on K^+^ currents mediated through the C-terminally deleted Kir6.2 channels (KirΔC26) and the effects of cGMP modulating agents on MB inhibition of cromakalim-activated K^+^ currents. (**A**) Currents from inside-out patches in response to two s-voltage-ramps from –100 to +100 mV under symmetric conditions (140 mM K^+^) were recorded in the absence and presence of 30 μM MB. Leak currents, recorded in the presence of 10 mM ATP, were subtracted. (**B**) Time-course of the effect of 30 µM MB on K^+^ currents mediated through the C-terminally deleted Kir6.2 channels (KirΔC26) in the absence (filled circles) and presence (open circles) of MB (30 μM), respectively. The horizontal bar represents the administration time of MB. (**C**) The effects of 20 min preincubation with BAY 58–2667 (3 µM) and 8-Br-cGMP (100 µM) on inhibition of cromakalim (100 μM)-activated K^+^ currents by MB (30 µM) in follicle-enclosed oocytes (*p* > 0.05, *n* = 5–7, Student’s *t*-test). MB: methylene blue; Bay 58: BAY 58–2667.

## Data Availability

Data will be available up on request.

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
