# Peer review of "Methylene Blue Inhibits Cromakalim-Activated K+ Currents in Follicle-Enclosed Oocytes"

_membranes, 2023, doi:10.3390/membranes13020121_

Round 1

Reviewer 1 Report

Methylene Blue (MB) has a pharmacologically-demonstrated vasoconstrictive effect. The results described in the present study provide strong evidence that cromakalim-activated K+ currents in follicular cells of Xenopus oocytes are directly modulated by MB.  MB inhibition of KATP channels may be one of the mechanisms contributing to the vasoconstrictive effects of MB but this possibility will need further investigation. The electrode voltage clamp techniques on follicle-enclosed oocytes are used expertly and the interpretation of data is convincing.

Specific comments.

- MB inhibits KATP channels by competing with the cromakalim binding site(s). The competition data suggest that  MB inhibits  K+-currents in a non-competitive manner. Does it mean that methylene blue, promethazine, chlorpromazine mefloquine, quinine, quinidine, and quinacrine share the same binding site.

-The statement that MB does not inhibit KATP channels by interacting with SUR, but rather by acting on the channel needs to be clarified or at least discussed.

Author Response

December 20, 2022

Re: Manuscript ID: membranes-2094094

Dear Editor,

Thank you for your review of our manuscript number membranes-2094094 entitled “Methylene Blue Inhibits Cromakalim-Activated K+ Currents in Follicle-Enclosed Oocytes” We appreciate the opportunity to address the concerns of the reviewers and have made the appropriate changes according to suggestions of reviewers as described below.

Reply to the comments of reviewer 1:

1) MB inhibits KATP channels by competing with the cromakalim binding site(s). The competition data suggest that  MB inhibits  K+-currents in a non-competitive manner. Does it mean that methylene blue, promethazine, chlorpromazine mefloquine, quinine, quinidine, and quinacrine share the same binding site.

REPLY: We thank the reviewer for pointing out this important issue. As the reviewer pointed out, phenothiazine-based molecules and structurally similar compounds such as promethazine [51], chlorpromazine [27,52] mefloquine, quinine, quinidine, and quinacrine [53,54] have been shown to directly inhibit KATP channels (citation numbers refer to the numbers provided in the manuscript reference format). Unfortunately, in these studies, the non-competitiveness of the phenothiazine effects has not been investigated. However, in an earlier study [53], phenothiazines including quinine, mefloquine, chloroquine, and quinidine, similar to our results with MB, were shown to inhibit C-terminally deleted form of Kir6.2 (KirΔC26) which does not contain sulphonylurea receptor; indicating that these compounds do not compete with sulphonylurea binding sites. Our radioligand bindings experiments with [3H]glibenclamide are also in agreement with their results. However, as mentioned above,  in the previous studies the issue of non-competitiveness was not investigated. Therefore, based on our results, we can only conclude that MB acts as a non-competitive inhibitor of K+ currents investigated in this study.

2) The statement that MB does not inhibit KATP channels by interacting with SUR, but rather by acting on the channel needs to be clarified or at least discussed

REPLY: We thank the reviewer for his/her valuable comments. In agreement with the suggestion of the reviewer, we further clarified this statement on page 9, lines 292-307, and incorporated the following statement:  

“Analysis of cromakalim concentration-response curves in the presence and absence of MB indicated that MB inhibits K+-currents in a non-competitive manner suggesting that MB does not act by inhibiting the cromakalim binding site on the channel. Similarly, radioligand binding studies with [3H]glibenclamide, a sulfonylurea-class antidiabetic drug, showed that MB does not significantly affect the glibenclamide binding site in follicle-enclosed oocyte membranes either. The pore of the KATP channel is formed from four Kir6.2 subunits, each of which is associated with a larger regulatory sulfonylurea receptor (SUR) subunit, which is the primary target for KATP blockers [43]. Importantly, MB also inhibited ion currents in oocytes expressing the C-terminally deleted mutant of Kir6.2 (KirΔC26), which forms functional channels but does not contain sulfonylurea binding site (SUR) [44]. Thus, the results of electrophysiological studies on KirΔC26 mutant channel and radioligand binding studies suggest that MB does not suppress the function of KATP channels by inhibiting known sulfonylurea binding site, but rather by acting at a different site on the KATP channel. Collectively, these results indicate that neither cromakalim nor sulfonylurea binding sites are involved in the inhibition of KATP channels by MB.”

Reviewer 2 Report

The article named Methylene Blue Inhibits Cromakalim-Activated K+ Currents in Follicle-Enclosed Oocytes is a well prepeared experimental paper utilized more electrophysiology based approach to membrane transport investigation. Experimental conditions have not be completely presented but are suitable for expert readers.

I ask authors to resolve some questions. At first, the dashed-lines of methylene blue application should be changed to dotted line.

Figure 3 presented data for the incubation time 60 min at 22 °C. Please explain the choice of incubation conditions for 60 minutes. If the dissociation constant is enough low a few minutes incubation could be sufficient for a specific binding. Long incubation followed to non-specific binding increase, since no one canceled the hydrogen exchange between molecules.

According to Figure 4B, MF cannot washed out after binding to the channel. Then the affinity with this channel should be very high. Please explain this experimental fact, which is not discussed further. In Figure 4C, the columns are labeled incorrectly. In all cases, MB was applied to the solution; therefore, BY 58+MB and 8-Br-cAMP+MB are more correct. Did I understand correctly or the caption to the picture is not clear.

line 265 = We have tested the involvement of guanylate cyclase activity by investigating the effects of BAY 58-2667 (3 μM).

Please explain how enzymes activity can be assessed in the inside out patch experiment when both the intracellular environment and almost all components except membrane-bound molecules are absent.

Author Response

December 20, 2022

Re: Manuscript ID: membranes-2094094

Dear Editor,

Thank you for your review of our manuscript number membranes-2094094 entitled “Methylene Blue Inhibits Cromakalim-Activated K+ Currents in Follicle-Enclosed Oocytes” We appreciate the opportunity to address the concerns of the reviewers and have made the appropriate changes according to suggestions of reviewers as described below.

Reply to the comments of reviewer 2:

1) I ask authors to resolve some questions. At first, the dashed-lines of methylene blue application should be changed to dotted line.

REPLY: We thank the reviewer for his/her valuable comments. In agreement with the reviewer’s suggestion, we have changed the dashed-line to the dotted line in Figure 1A.

2) Figure 3 presented data for the incubation time 60 min at 22 °C. Please explain the choice of incubation conditions for 60 minutes. If the dissociation constant is enough low a few minutes incubation could be sufficient for a specific binding. Long incubation followed to non-specific binding increase, since no one canceled the hydrogen exchange between molecules.

REPLY: We thank the reviewer for his/her valuable comments. Our radioligand binding assay protocols were developed by Dr. Zakharova using the trial and error method [cited as reference 32 in the manuscript]. Glibenclamide is a highly lipophilic agent and Dr. Zakharova found that, at low concentrations (1-10 nM) used in radioligand binding studies, glibenclamide requires long incubation times to achieve better specific to non-specific binding ratio. Specifically, we have tested 15, 30, and 60 min incubation times, and a higher specific to non-specific binding ratio was observed at 60 min incubation time. We have used this assay protocol earlier and published our results as stated in the methods section [cited as reference 33 in the manuscript].

3) According to Figure 4B, MF cannot washed out after binding to the channel. Then the affinity with this channel should be very high. Please explain this experimental fact, which is not discussed further. 

REPLY: We thank the reviewer for his/her comments. We agree with the reviewer in that the effect of MB on K-ATP channels does recover only partially (about 25% after 40 min washout period in Figure 1B and 20% after 1 min in Figure 4B). In agreement with our results, the effects of several phenothiazines including chlorpromazine [cited as reference 52 in the manuscript] as well as quinine,  mefloquine, and chloroquine [cited as reference 53 in the manuscript] do not recover within 1 to 5 min washout times. We do not know why MB has only partial recovery of its effect on K-ATP channels. However, MB has also been shown to interact with lipid membranes (Schmidt et al., 2015; Ileri Ercan et al., 2018)

Schmidt, T. F., Caseli, L., Oliveira, O. N., Jr, & Itri, R. (2015). Binding of methylene blue onto Langmuir monolayers representing cell membranes may explain its efficiency as photosensitizer in photodynamic therapy. Langmuir : the ACS journal of surfaces and colloids, 31(14), 4205–4212. https://doi.org/10.1021/acs.langmuir.5b00166     

Ileri Ercan, N., Stroeve, P., Tringe, J. W., & Faller, R. (2018). Molecular Dynamics Modeling of Methylene Blue-DOPC Lipid Bilayer Interactions. Langmuir : the ACS journal of surfaces and colloids, 34(14), 4314–4323. https://doi.org/10.1021/acs.langmuir.8b00372   

4) In Figure 4C, the columns are labeled incorrectly. In all cases, MB was applied to the solution; therefore, BY 58+MB and 8-Br-cAMP+MB are more correct. Did I understand correctly or the caption to the picture is not clear.

REPLY: We thank the reviewer for his/her comments. In agreement with the reviewer’s suggestion, in this version of the manuscript, we modified the labels in Figure 4C and included new labels as BAY-58 + MB and 8-Br-cAMP + MB.

5) line 265 = We have tested the involvement of guanylate cyclase activity by investigating the effects of BAY 58-2667 (3 μM). Please explain how enzymes activity can be assessed in the inside out patch experiment when both the intracellular environment and almost all components except membrane-bound molecules are absent.

REPLY: We thank the reviewer for his/her valuable comments. We apologize for the confusion regarding the experimental methods related to Figure 4C. These experiments were conducted using the two-electrode voltage clamping method in follicle-enclosed oocytes. The inside-out patch clamp technique was not used in these experiments. In this revised version of the manuscript, we clarified this matter by stating “in the follicle-enclosed oocytes” on page 9 line 263.

Round 2

Reviewer 2 Report

No comments.